# Visual Speech Recognition with Lightweight Psychologically Motivated Gabor Features

**DOI:** 10.3390/e22121367

**Published:** 2020-12-03

**Authors:** Xuejie Zhang, Yan Xu, Andrew K. Abel, Leslie S. Smith, Roger Watt, Amir Hussain, Chengxiang Gao

**Affiliations:** 1Department of Computer Science and Software Engineering, Xi’an Jiaotong-Liverpool University, Suzhou 215123, China; xuejie.zhang17@alumni.xjtlu.edu.cn (X.Z.); yan.xu@xjtlu.edu.cn (Y.X.); chengxiang.gao16@alumni.xjtlu.edu.cn (C.G.); 2Faculty of Natural Sciences, University of Stirling, Stirling FK9 4AL, UK; lss@cs.stir.ac.uk (L.S.S.); r.j.watt@stir.ac.uk (R.W.); 3School of Computing, Edinburgh Napier University, Edinburgh EH11 4DY, UK; a.hussain@napier.ac.uk

**Keywords:** speech recognition, image processing, gabor features, lip reading, explainable

## Abstract

Extraction of relevant lip features is of continuing interest in the visual speech domain. Using end-to-end feature extraction can produce good results, but at the cost of the results being difficult for humans to comprehend and relate to. We present a new, lightweight feature extraction approach, motivated by human-centric glimpse-based psychological research into facial barcodes, and demonstrate that these simple, easy to extract 3D geometric features (produced using Gabor-based image patches), can successfully be used for speech recognition with LSTM-based machine learning. This approach can successfully extract low dimensionality lip parameters with a minimum of processing. One key difference between using these Gabor-based features and using other features such as traditional DCT, or the current fashion for CNN features is that these are human-centric features that can be visualised and analysed by humans. This means that it is easier to explain and visualise the results. They can also be used for reliable speech recognition, as demonstrated using the Grid corpus. Results for overlapping speakers using our lightweight system gave a recognition rate of over 82%, which compares well to less explainable features in the literature.

## 1. Introduction

The brain handles both auditory and visual speech information. Visual information was shown to be able to influence interpretation, as demonstrated by the McGurk Effect [1]. Lipreading aims to recognize speech by interpreting lip movement [2], and is a technique that has always been used by people with hearing loss [3]. Vision also supplements audio under adverse acoustical conditions [4]. Lipreading is widely used in speech recognition, identity recognition, human-computer interfacing and multimedia systems, and traditionally, it has two key components: lip feature extraction and feature recognition (front-end and back-end). However, some lipreading systems use deep learning methods, and are end-to-end systems not separated into two stages, instead relying on data-intensive pre-trained models, and lacking clearly defined explanatory features. Many systems that use pre-trained deep learning models require very data intensive models that can be extremely time consuming to train [5].

Another issue is the lack of explainability in these systems. The end-to-end approach uses the image directly as input, with no feature extraction, making it very challenging to visualise and explain the features. Although recent research investigated the way that Convolutional Neural Networks (CNNs) self-learn features [6], deep learning systems remain very hard to explain. The alternative two stage approach, which extracts features using more conventional image techniques such as Principal Component Analysis (PCA) or Discrete Cosine Transform (DCT) [7] means that the features are generally transformed to different dimensions, and are not intuitive to humans when inspected. Another approach is to use shape or appearance models [8], which can adapt to fit mouth regions, and from these, geometric features can be identified. Again, these can be time consuming to train. Ideally, features should be quick to extract, should be robust enough to apply to new speakers without training, should be usable for machine learning, and crucially, should be explainable and intuitive so that we can understand the network’s behaviour.

Another key motivation is to extract simple and lightweight features. Many recent systems use deep learning CNN approaches [9,10] that use an original image or sequence of images as an input into the system. However, in a real world situation, data may need to be extracted and transmitted wirelessly in real time. One example of this is with wireless hearing aids [11], where hearing aids are linked by bluetooth [12] or by radio to carry out noise reduction or source separation by linking microphones from both hearing aids, or even from external sources [11]. This requires rapid real time transmission of data, as well as low power devices. The data processing needs to be extracted quickly, and the features should be low dimensionality. Previous work by the authors focused on the potential of developing noise filtering systems using visual information [13,14], and in the case of proposed systems such as this, data would need to be collected by a camera, transmitted wirelessly, quickly and accurately, and then processed in order to be able to produce a real time output. As many researchers are currently developing image-based speech processing systems [15,16], inspired by the role of vision in human hearing [1], it is not infeasible that a camera may become part of a future hearing aid system. In this scenario, being able to extract accurate, functional, and lightweight features becomes very important.

Motivated by psychological research into how humans recognise faces, we present Gabor-based lip feature extraction. These features are quick and easy to extract and use. This is helpful for training with limited data, or for developing more lightweight speech recognition systems. In contrast to other image features such as DCT, the extracted features can be clearly visualised and interpreted over time. These are a form of geometric features, and further, as well as the more conventional two dimensional height and width, we are also able to extract three dimensional features to visualise the depth of the mouth opening. Although we are working with two dimensional images, by identifying the mass of the mouth opening, we are able to distinguish between different types of mouth openings such as fully open mouths and gritted teeth, meaning that we can extract 3D mouth information. This paper extends an initial conference paper presented at IEEE SSCI 2019 [17] by demonstrating that as well as being able to visualise the features, we can also use these simple and lightweight features for visual speech recognition.

In this paper, we present a simple, quick, and reliable three dimensional feature extraction method, using Gabor filtering to identify the lip region and extract simple, explainable and visualisable parameters. We then use our feature extraction method to conduct quick, simple and efficient machine learning with bi-directional Long-Short Term Memory (LSTM) networks for speech recognition. The results show that our features can be successfully used to recognise speech from the Grid corpus, using low-dimensionality features and a LSTM-based network. We experimented with various configurations and identified that optimal results could be achieved using a Bidirectional LSTM model with 6 hidden layers. These features are much simpler and quicker than using a deep CNN type model, while still returning good results. We also show that the inputs into the network (i.e., our features) can demonstrate consistent temporal patterns, making it much easier to explain them to human observers.

Section 2 provides a detailed background of relevant research, and is followed by an introduction to our psychologically motivated Gabor features in Section 3. The detailed feature extraction approach is presented in Section 4 and the network and dataset configuration is presented in Section 5, followed by a discussion on parameter selection in Section 6. We present a brief individual word analysis in Section 7, and detailed speech recognition results are presented and discussed in Section 8 and discussed in Section 9, showing that very good lipreading results can be achieved using simple features.

## 2. Background

Table 1 shows several examples of state-of-the-art machine learning lipreading methods, based on [18], extended with more recent research. Relevant examples are discussed in more detail in Table 1.

As well as a variety of network topologies, datasets, and tasks, the different approaches summarised in Table 1 also have different training and test conditions. Several approaches use overlapping speakers (i.e., training and testing with the same speakers): Assael et al. [15], Wand et al. [19], Grid corpus experiments by Chung et al. [16], Wand et al. [20] and Xu et al. [21]. Others use unseen speakers (i.e., speakers that the system has not been trained with): Chung and Zisserman [22], OuluVS2 corpus experiments in Chung and Zisserman [23,24], Petridis et al. [25] and Fung and Mak [9]. Finally, other research uses the BBC program based corpora, meaning that the training and test sets are divided according to broadcast date. This means that there may be some speaker overlap, depending on the dataset. This includes research by Chung and Zisserman [23,24], Chung et al. [16], Stafylakis and Tzimiropoulos [26], Petridis et al. [10] and Weng [27]. This demonstrates that as well as different techniques and corpora, the training and test conditions also vary, making direct comparisons very difficult.

The GRID corpus [28] is widely used for research. Assael et al. in 2016 proposed an original architecture LipNet [15], which achieved a very high sentence-level recognition rate of 95.2%. It uses spatiotemporal convolutional neural networks (STCNNs), recurrent neural networks (RNNs), and the connectionist temporal classification loss (CTC). Due to the model being end-to-end, it does not need feature extraction, and the individual word alignments in the database are not needed, as it is used at sentence level. However, there are limitations with this approach. Faisal and Manzoor identified that this system is not appropriate for Urdu, as the output always consists of 6 words, regardless of input. This results from Lip-Net being trained with Grid at the sentence level, thus fixing the output form [29]. Wang et al. [19] used an LSTM for lipreading with a two-layered neural network on the Grid corpus. They compared the LSTM with two different methods: Eigenlips and Support Vector Machines(SVM), and Histograms of oriented Gradients (HOG) and SVM. LSTM gave better word accuracy results, 79.5%.

Chung et al. [16] proposed a new architecture called Watch Listen Attend and Spell (WLAS) at character-level, which uses a very large English language dataset called Lip Reading Sentences (LRS), based on BBC broadcasts. They also produce good results on the Grid corpus with a Word Error Rate(WER) of only 3%, with auditory information also used to enhance the result. This result is very similar to that obtained by Xu et al. in 2018 [21]. They proposed a new architecture called LCANet, which includes 3D-convolution, highway network, Bi-GRU and attention-CTC networks, this architecture had a WER of only 2.9%, although was focused on its specific corpus, and again, involved an intensive CNN trained with a lot of data. Xu et al. and Assael et al. [15] used a 3D-CNN differing from the CNN that Chung et al. used, as it captures temporal information. LCANet considered the degradation of the deep learning neural network, and a Highway network is an alternative design that provides a simple and effective way to optimise deep-layer neural networks [30]. Some other architectures also take this problem into consideration. ResNet was used to deal with this problem [10,26]. However, approaches such as these rely on very data-intensive pre-trained models. These are often limited in their wider application potential and lack clearly defined features that can be used for explanation and to enable improved human understanding. We wish to present features that have a clear psychological motivation, and are also quick and easy to use.

The word recognition rate of the GRID corpus is often much higher than other corpora such as OuluVS2, LRW, and LRS. However, with constant strengthening of the different deep learning neural network models, these also achieved low WERs. Martinez et al. [27] achieved a 85.3% word recognition rate for the LRW corpus by using a residual network and a Bidirectional Gated Recurrent Unit (BGRU) [31]. For OuluVS2, Petridis et al. [25] obtained a high recognition rate (94.7%). Clearly, there are a wide range of results reported in the literature, with some reporting extremely good results. However, although the NNs discussed above have a high recognition rate for characters, words, or sentences in different corpora, all of them use images of the lip region as input rather than lip features. It is not easy for researchers to explain, using these features, how lip features are discriminated, and thus they spend more time on model training. It should be noted that these results are hard to generalise. They are an example of solving a problem for a specific corpus, as in the issues found by Faisal and Manzoor. Rather than attempting to gain a slight improvement on the Grid Corpus, we wished to develop fast and lightweight lip feature extraction, and combine it with a relatively simple model to show that good results could be achieved using a simpler and more explainable approach than the time and data intensive CNN approach.

Classical methods can be used to extract lip features, and are arguably more lightweight and explainable than CNNs [13]. A variety of approaches were proposed, with some examples shown in Table 2. This table lists several key techniques used for feature extraction, as well as which classifier was used, and how they were evaluated (database, task, and recognition rate). It should be noted that the main focus here is on techniques, rather than recognition results.

Again, as discussed previously, different approaches also have different training and testing conditions. Of the approaches summarised in Table 2, several researchers use overlapping speakers including: Shao and Barker [33], AVLetters corpus experiments by Zhao et al. [35] and Cappelletta and Harte [39]. Seymour et al. [34], OuluVS corpus experiments in Zhao et al. [35], Lan et al. [36] and Lee et al. [40] achieved their results using unseen (i.e., non overlapping train/test) speakers.

There are two main kinds of feature extraction: model-based methods and pixel-based methods [41]. Model-based methods locate the contour of the mouth, and obtain the width, height, etc. of the mouth. Examples of this include active appearance models (AAM) [8], and active contour models (ACM) [32]. These describe the mouth contour by using a set of key feature points. Pahor et. al [42] noted that these methods are computationally expensive due to the mouth model deformation, and the model definition needs the prior knowledge of some features of the lip image, meaning that with novel or unexpected data, they can perform badly. However, they do have the benefit of being possible to visualise and explain.

Pixel-based methods use the gray image or feature vectors after pre-processing of the lip image. Examples include the Discrete Cosine Transform (DCT), Gabor Wavelet Transform (GWT), Principal component analysis (PCA), linear discriminant analysis (LDA), optical flow and LBP from Three Orthogonal Planes (LBP-TOP). Bhadu et. al pointed out that DCT is good at concentrating energy into lower order coefficients, but that it is sensitive to changes of illumination [43]. This is not always an issue, and other speech processing research successfully used DCT features for speech processing research with image data [7,44,45]. However, in previous work, Abel et. al argued that DCT features are difficult to explain and analyse, because they consist of applying a frequency domain transform then ordering components. This means that the resulting features are difficult to explain to the user and are not easy to visualise in an intuitive manner [17]. PCA can minimize the loss of information and does not require a clear contour. However, the results are similar to DCT and LDA, and PCA is sensitive to illumination [46]. For PCA, the result of positioning and tracking is difficult to test because of the non-intuitive intermediate processing result [41], making visualisation difficult.

Zhao et al. [35] proposed the use of LBP-TOP to extract features. However, Bharadwaj et al. [47] considered that LBP-TOP is computationally expensive making real-time applications difficult. Optical flow is a widely used method that can extract lip motion parameters and can analyse motion, but it requires accurate positioning at the pre-processing stage [41]. In this case, Gabor transforms are insensitive to variation in illumination, rotation, scale, and can be used to focus on the facial features such as eyes, mouth and nose, and have optimal localization properties in both spatial and frequency domains [43]. Overall, when it comes to lipreading systems, the features tend to be either CNN-based inputs (representing neuron weights and thus not easily visualisable, and requiring heavy training), pixel-based approaches (not requiring so much training, but tending to have a non-intuitive visualisation), or model based methods (intuitive and explainable, but requiring heavy training, and not coping well with novel data).

## 3. Psychologically Motivated Gabor Features

Humans recognise faces using distinctive facial features. Independent facial perceptual attributes can be encoded using the concept of face space [37,48], where distinctiveness is encoded as the difference from an overall average. The biological approach to face recognition provides evidence that humans use early-stage image processing, such as edges and lines [49]. Dakin and Watt [37] used different Gabor filter orientations, identifying that horizontal features were the most informative, and that distinct facial features could be robustly detected. This was developed further by [17,50]. The coarse distinctions between facial features can also be applied to more finely detailed features [37], with clear differences between features such as the lips, teeth, philtrum, and mentolabial sulcus. This enables quick and accurate mouth feature information to be obtained, with a three dimensional representation of the mouth opening possible (i.e., tracking the width, height, and also using the colour information to identify the depth of the mouth opening). Thus, the principle of horizontal Gabor features can also be applied to lip specific feature extraction to generate human-centric features.

The impulse response of a Gabor filter is defined as a sine wave (sine plane wave for a 2-D Gabor filter) multiplied by a Gaussian function. The filter is composed of a real and imaginary part, which are orthogonal to each other. The complex form is:(1)$$g\left(x,y;\lambda ,\theta ,\psi ,\sigma ,\gamma \right)=\exp\left(-\frac{x^{\prime }{}^{2}+\gamma^{2}y^{\prime }{}^{2}}{2\sigma^{2}}\right)\exp\left(i\left(2\pi \frac{x^{\prime }}{\lambda }+\psi \right)\right)$$

Sujatha and Santhanam [51] used GWT to correct the mouth openness after using the height-width model when extracting lip features. These two features are used as input into an Ergodic Hidden Markov model for word recognition, with 66.83% accuracy. They used GWT as a tool to correct the mouth openness, and only used two 2D lip features to recognize words. Hursig et al. also used Gabor features to detect the lip region [38]. However, in this research, seven lip features are extracted, including six 2D features and one 3D feature, as we obtain and identify detailed lip features rather than the overall lip regions. This is relevant, because we can produce time domain vectors of these features, and these can be measured and visualised.

## 4. Proposed Feature Extraction Approach

### 4.1. ROI Identification and Tracking

Figure 1 shows the key components of the system. Given an image sequence $I_{n}\left(n=1\ldots N\right)$ from a video file, our aim is to track the lip region. For ROI identification, we follow previous research [7,13] and use a Viola-Jones detector and an online shape model [52], similar to previous Gabor lip feature research [38]. This outputs a coarse 2-D lip region for each image frame, represented as the four (x,y) coordinate pairs $(L_{x}^{n}\left(i\right),L_{y}^{n}\left(i\right)),i=1\ldots 4$. These can identify $C_{L}^{n}$, the ROI centre point for each frame.

### 4.2. Gabor Feature Generation

Following Dakin and Watt [37], we calculate horizontal Gabor features with a Fast Fourier Transform (FFT). This generates positive and negative going real and imaginary components, and we use the real component. Each image is converted to greyscale, and Gabor filtering is applied, see Figure 2b. To reduce small values such as background noise and regulate the size of the image patches, a threshold is applied to the initial transform, (see Figure 2c). Several parameters are required, which generally only need adjusted when a different corpus is used, or a speaker is sitting at a prominent angle, or at a different distance from the camera:-Gabor wavelength λ: this can feasibly be between 2 and 20+. The exact parameter depends on image size.-Filtering threshold *t*: used to ignore minor face features and background noise. The range is 0 to 1, and a value between 0.05 and 0.3 was found to be effective.-Face angle orientation Θ (degrees): if the speaker has their head straight, then 0 (horizontal) is suitable, but a slight angle, commonly Θ=5, may be needed.-Minimum region patch area $P_{\mathrm{MIN}}$: this is useful when speakers have (for example) a prominent chin or teeth. A value of between 50 and 100 is suitable.

### 4.3. Image Region Patches and Relevant Patch Identification

After filtering and thresholding, the most prominent regions (i.e., the local extrema in the filter outputs) are calculated, and these regions are grouped and represented by rectangular image patches. These are calculated using the filtered real component of the transformed image, shown in Figure 2c.

Given a filtered image, patches are then created. Using the Matlab “bwconncomp” function, with 8 degrees of connectivity, and the “regionprops” function, *R* groups of connected pixels are created. The result is a matrix of pixel locations, $Q^{X}$ and $Q^{Y}$, and values $Q^{V}$ for each grouping, $G_{r}$. For each $G_{r}$, the area $A_{r}$ is defined as the number of pixels in each $G_{r}$. The mass is calculated by summing the pixel values, $M_{r}=\sum_{p=1}^{P}Q_{p}^{V}$.

The patch centre coordinates, $X_{r}$ and $Y_{r}$ are calculated using both pixel coordinates $\left(p_{x},p_{y}\right)$ and pixel values. As some of the pixels at region edges are not as strongly connected, this is taken into account,
(2)$$X_{r}=\sum_{p=1}^{A_{r}}\left(p_{x}*Q_{p}^{V}\right)/M_{r}\;\mathrm{and}\;Y_{r}=\sum_{p=1}^{A_{r}}\left(p_{y}*Q_{p}^{V}\right)/M_{r}$$

The variance is calculated, $\sigma_{X_{r}}^{2},\sigma_{Y_{r}}^{2}$, as is the covariance,
(3)$$\sigma \left(X_{r},Y_{r}\right)=\sum_{p=1}^{A_{r}}\left(p_{x}*p_{y}*Q_{p}^{V}\right)/M_{r}-X_{r}*Y_{r}$$

The patches are not always horizontal or vertical, and have an orientation, Θ, calculated using covariance and variance,
(4)$$\Theta =\tan^{-1}(2*\sigma \left(X_{r},Y_{r}\right),\left(\sigma_{X_{r}}^{2}-\sigma_{Y_{r}}^{2}\right))/2$$

Θ can then be used to calculate width and height of each patch. The width is calculated as,
(5)$$W_{r}=\sqrt{\left(|w_{r}|+0.5/\pi \right)}$$
where $w_{r}=\left(X_{r}^{2}-{\left(X_{r}\right)}^{2}\right)*\cos^{2}\Theta +2*\sigma \left(X_{r},Y_{r}\right).*\left(\cos\Theta *\sin\Theta \right)+\sigma_{Y_{r}}^{2}*\left(\sin^{2}\Theta \right))$. This also requires the squared value,
(6)$$X_{r}^{2}=\sum_{p=1}^{A_{r}}\left(p_{x}^{2}Q_{p}^{V}\right)/M_{r}$$

The height, $H_{r}$ is calculated with a similar process,
(7)$$H_{r}=\sqrt{\left(|h_{r}|+0.5/\pi \right)}$$
where $h_{r}=\left(Y_{r}^{2}-{\left(Y_{r}\right)}^{2}\right)*\cos^{2}\Theta -2*\sigma \left(X_{r},Y_{r}\right).*\left(\cos\Theta *\sin\Theta \right)+\sigma_{X_{r}}^{2}*\left(\sin^{2}\Theta \right))$ with
(8)$$Y_{r}^{2}=\sum_{p=1}^{A_{r}}\left(p_{y}^{2}Q_{p}^{V}\right)/M_{r}$$

These properties allow for the creation of patches. An example is shown in Figure 2d, showing all the resulting patches generated from this image. There are patches generated around the hair, eyes, nose, mouth and shoulders. We are most interested in the mouth opening patch. For each patch, several values are generated. These are quick to generate, and can be used for analysis. The most relevant are:**Width** The width of the lip region, $W_{r}$**Area** The height is constrained by λ, but $A_{r}$ can be a good measure of mouth opening.**Mass** Mass ($M_{r}$) is related to intensity. It shows the mouth depth, providing 3D representation, and can distinguish between a closed mouth, an open mouth showing teeth, and a fully open mouth.**Xpos** The *x* position, $X_{r}$ identifies the mean x-position of the pixels in the patch, i.e., the centre position of the x-co-ordinate.**Ypos** The *y* position, $Y_{r}$ identifies the mean y-position of the pixels in the patch, i.e., the centre position of the y-co-ordinate.Θ 
This is used to calculate the orientation of each patch. This differs from the orientation of the Gabor wave Θ: Θ corresponds to each patch orientation, so for example, each shoulder in Figure 2d would have a different orientation.

To calculate the ROI centre point, $C_{L}^{n}\left(x,y\right)$, is calculated. This can be used to identify the lip region patch. For each frame, the ROI centre point, $C_{L}^{n}\left(x,y\right)$ is compared to each *r*-th object of *X* and *Y* for each *n*-th frame to identify the closest patch, which is defined as the mouth opening patch. This is shown in Figure 2f, showing the ROI as a pink rectangle, and then the chosen patch (the mouth opening) in blue. The complete process is shown in Figure 2.

### 4.4. Extracted Features

The output can be visualised as a sequence of frames, showing the ROI and the lip features. Figure 3 shows an example frame from the Grid Corpus [28]. Both the mouth dimensions and the mass can be calculated quickly. We visualise this with a lighter blue being used for an open mouth, and darker for a closed mouth. Figure 3a shows an open mouth, with a large box and dark colour, Figure 3b shows an open mouth, but without depth (due to the teeth being visible), reflected by the lighter colour. This represents the three dimensional aspect of the features, in that it can make depth distinctions on 2D images. Finally Figure 3c shows a closed mouth. These outputs can also be visualised as vectors, showing that different properties can be clearly and simply visualised over time. This will be discussed in Section 7.

## 5. Word Recognition with Bidirectional LSTM Neural Networks

### 5.1. System Configuration

All machine learning and feature extraction took place on a desktop machine, with Windows 10 Pro installed. The CPU was an Intel i7-8700K with a 3.70 GHz clock speed, and 32 GB of RAM. The machine had a NVIDIA GeForce GTX 1080 Ti, although our calculations used the CPU. All experiments were carried out using Matlab 2018.

### 5.2. Dataset

We used the widely used GRID audiovisual database [28] for visual speech recognition (see Table 1 and Table 2). It contains 34 speakers, with 1000 video files for each. Rather than simply using a big data approach and training the entire database, we wanted to investigate the performance on individual speakers, and so we chose to focus on a subset of speakers. To do this, we created a balanced dataset of the most commonly used individual words in the dataset. The chosen words and their distribution for a single speaker is shown in Figure 4.

This differs from end-to-end lip reading approaches, which use very large pre-trained models, further trained using entire speech databases. We therefore focused on speakers S1, S2, S3, S4, S5, S6, S12, S13, S14, S15, S16, S17, S18, S19, and S20, with training and validation sets randomly selected. As with other work in the literature using this corpus, we are able to use the alignment data, which labels the start and the end of each word. It is possible to use speech segmentation software to split the words automatically [53], but for this research, the labelling is sufficient.

### 5.3. Gabor Feature Preprocessing

As discussed previously, extracted features include several properties. Individual values vary between speakers: for example, some speakers may be closer to the camera than others, so we normalise the features between 0 and 1 so that they have equal weight scales.

As well as the key features discussed in Section 3, width, area, Xpos, Ypos, and Θ, we use additional information. This includes the centre points of the lip region, the height (of limited utility in the current implementation, where height is fixed to Gabor wavelength), the patch id (a label identifying its location), the elongation (an extension applied for visualisation), and the amplitude, which is a measure closely related to mass, for a total of 11 features. While not vital for visualisation, here we use them to provide additional input for machine learning. However, due to the use of the tracker, the precise *x* and *y* co-ordinates of the mouth region may contain potential noise data due to very small fluctuations. The centre points of the data tend to be very stable, so we therefore use the dynamic amplitude coordinates (x,y) of the lips and subtract the central coordinates (cx,cy) to obtain more accurate visual features.

Another aspect to consider is the feature extraction time. The steps discussed in Section 4 are fairly quick, with the Gabor feature extraction itself being very quick. The implementation for the Viola-Jones detector is a little more time consuming, which slows the feature extraction down. This means that for a 2 s video, the extraction currently takes around 10–15 s, on the computer used here. However, this is not included in the training time discussed later in this paper, as we extracted all features from the dataset in a batch implementation.

### 5.4. Bidirectional LSTM Model and Optimization Criterion

In the experiments reported here, we used bidirectional LSTM machine learning [54]. The temporal nature of the data was well suited to the use of LSTMs [55]. Preliminary experiments showed that using unidirectional LSTMs worked well for single speaker recognition but were not satisfactory for multiple speaker models, whereas bidirectional LSTMs were more satisfactory.

We investigated the use of seven similar neural network models, varying the number of hidden layers from 3 and 9. All models use bidirectional LSTMs, and each layer contains different numbers of nodes, as shown in Table 3, and represented in Figure 5 For classification, there is a fully connected layer which matches the number of classes, and a classification layer (which computes the cross entropy loss).

Another factor affecting result is the parameter adjustment of the training model. Based on the characteristics of the stochastic learning gradient algorithm [56,57], we performed preliminary experiments using RMSProp, Adam, and AdaGrad learning methods. We used RMSProp as it performed best. Secondly, since the length of each word is different, we chose the longest sequence length of the words as the computing length during each epoch training stage.

To calculate the training time, we trained the data with a single speaker from the GRID database, S3, and calculated the average training time, as shown in Table 4. This table shows that a 3 layer model has a mean training time of 41.75 min, increasing to 62.95 min for 4 layers, 78.40 min for five layers, increasing at a fairly linear rate. This shows that a 6 layer model can be trained with data from 800 different videos from a single speaker relatively quickly.

## 6. Tracking and Parameter Selection

In preliminary experiments, we successfully tracked thousands of videos from multiple corpora, including Grid [28], and VidTIMIT [58]. These were chosen due to their wide use in speech processing research. The tracker was found to be effective for our needs, although it could be replaced by other approaches. We used a Viola-Jones detector, with a shape tracker as used in previous research [7,13]. Although it is possible to upgrade this approach, we found that the Viola-Jones detector was suitable for the relatively stable Grid corpus. Parameters were kept as consistent as possible, with only slight adjustments. In almost all cases, the Gabor wavelength λ was set to 5, with a slightly larger λ for higher resolution frames, and in almost all cases, the patch area was set to 50. The threshold *t* varied between 0.14 and 0.25 depending on experimentation, and Θ was generally set to 0, although setting it to 5 is useful when the speaker is at a slight angle. Here, all features were extracted from the video file without any offline training being required, although a small number of videos had their parameters adjusted and were re-run. We found that although these hyper-parameters required this initial tuning, once they were tuned, they were robust across different corpora. The key parameter that needed to be changed was the wavelength, which had to change depending on the size of the image, but otherwise, the parameters were suitable for all cases that we tried, meaning that our hyper-parameters are relatively stable, with minimal adjustments needed.

## 7. Individual Word Analysis Results

This section demonstrates feature visualisation. Detailed results were presented in a previous conference paper [17], and we provide a brief summary here. We aim to produce simple data that can demonstrate word relationships, and extracted several example sentences from the Grid corpus. In the figures in this section, the *x*-axes correspond to the number of frames, with one data point for each frame, with the *y*-axes representing amplitude. The amplitude changes are of interest, as they show the differences between individual frames, and also between content and speakers. We refer to these as explainable features, because they provide a simple time domain representation of speech. We can see how the properties of the mouth, (width and area) change over time, as well as having a 3-D element. By being able to see not just the 2-D area of the mouth, but by calculating the mass, we can also measure how open the mouth is using colour changes. This allows us to distinguish between an open mouth with the teeth together, and an open mouth with the teeth apart.

As speakers have different speech rates and mouth sizes, we normalise over time and amplitude, and use word alignment data (provided in the Grid corpus) to identify individual words. The manual alignment data is not fully precise, so for visualisation purposes, we adjust the *x*-axes slightly where appropriate to match peaks. A simple example of this is shown in Figure 6, where the word ‘please’ is said by 10 different speakers. The mass peaks when the mouth opens during pre-voicing, followed by a closing for the plosive of ‘p’. The area and width are very similar: only the area is shown here, with the narrowest area before the ‘p’ is formed, which expands around the ‘ee’ stage, before closing slightly for the ‘s’ part. Again, all 10 sentences here show a similar pattern for all 10 speakers. Finally, we plotted 6 different words from the same speaker in Figure 6 (bottom), showing ‘at’, ‘bin’, ‘blue’, ‘now’, ‘nine’, and ‘q’. Despite the normalisation, there is no clear pattern, showing that our approach can identify individual words very simply. To demonstrate the effectiveness of this approach, we present the results of speech recognition work and compare them.

These examples show that we can visualise our vectors, and as well as individual words we can easily track the key parameters over time as shown in Figure 7. This means that we can identify both key visual differences and similarities between different words. Thus, when lipreading systems either perform well or fail, we can explain why this is the case.

## 8. Speech Recognition Results

The speech recognition experiments are the key contribution of this paper, and are divided into three parts: single-person models, two-person models and multiple-person models. The mean of five runs of randomised training and validation sets was used to reflect the overall value, and the Interquartile Range (IQR) used to describe the dispersion and stability of data. All models followed the bidirectional LSTM approach presented in Section 5. While we can compare our results to those reported in Table 1, and we generated comparable results, the key contribution is to demonstrate that we can generate these results with human-centric and explainable features (as opposed to other less intuitive features such as DCT or CNN features).

### 8.1. Single-Person Model

Single person models are models where the speaker is trained on a single speaker, then tested with new sentences from the same speaker, so that although the sentences are unseen, the speaker is known. This allows us to evaluate how different models (i.e., with different numbers of hidden layers) perform. In the initial experiments, we trained models on speaker S2 and S3 from Grid. We trained using the models described in Section 5 with different numbers of hidden layers. For these models, the validation dataset accounts for 20% and the training dataset accounts for 80%. The results are shown in Figure 8 and in Table 5.

As shown in Figure 8, the average for S2 is 83%, and for S3 is 80%. S2 has a slightly higher result, but the IQR is low for both speakers. In Figure 8, abnormal results can be seen in the nine layer model for the S2 dataset, the IQR is larger than other layers, and the average is lower than other models for both speakers. This suggests a possible lack of training data. As a result, and as we are not taking a big data approach, we focus on using smaller models.

We also trained single speaker models using S4 and S5 datasets, using a ratio of training data to validation data of 70/30. The recognition rate of the same model for the S4 dataset is about 86.72%, as shown in Table 6. The IQR of S4 and S5 is 0.0076 and 0.01 respectively, meaning that the fluctuation of the result is small. Finally, Figure 9 shows the confusion matrix diagram for the S5 validation dataset for one set of tests. The accuracy is very good, except for ‘red’ and ‘soon’. This is a surprising mix up, because the two words both sound and look different. One similarity is that the words can have similar mouth shapes and movements when pronouncing them. The recognition accuracy of S5 is about 86.21%, (detail shown in Table 6). These experiments showed that focusing on a small model with single speaker training could achieve consistently reliable results.

### 8.2. Two-Person Model

From the experiments with a single speaker model, we identified that using a 6 layer model was stable and provided good results. We therefore trained a dual speaker model, using data from GRID speakers S3 and S4. The training/validation split was 65/35. Here, we wished to evaluate if our simple model and feature extraction approach could work reliably with more than one speaker. The results are shown in Table 7. These show that the accuracy of S3 is 80.52% while the recognition rate of S4 is 85.52% after five runs, with very consistent results. In comparison to single-person models, the average recognition rate of S4 is a little lower, while the average recognition rate of S3 is higher, but overall it shows that a single speaker model can be successfully extended to more than one speaker.

### 8.3. Multi-Person Model

Finally, we also experimented with a multi-speaker model, training a 6 layer model with data from 6 speakers, with the aim of assessing how well a larger model handles overlapping speakers, and how well it can generalise to new speakers. The model was trained using data from GRID speakers S2, S3, S4, S5, S6, S15. The results from overlapping speakers can be seen in Figure 10 and Table 8.

For the overlapping speakers, the recognition accuracy ranges from 78% to 86%. The mean recognition rate is 82.82% with the 6 layer model. This shows that using a simple feature extraction method, combined with a bidirectional LSTM model, can achieve results comparable to those in the literature which use more complex and time consuming methods. We also investigate how well the system was able to generalise to new speakers, by using this trained model with validation from GRID speakers S1, S12, S13, S14, S16, S17, S18, S19, and S20. The results are shown in Figure 11 and in Table 9.

For the unseen speakers, the average recognition rate is 50.32%. There are distinct differences between individual speakers, with some speakers performing at over 60% accuracy, while others have a much lower accuracy of 34%. This is not unexpected: we did not expect that the system would be able to fully generalise. However we note that for several speakers, the results are surprisingly good, using a very simple feature extraction technique and machine learning model.

To analyse the results in more detail, we provided the confusion matrices of the validation datasets for several speakers. Figure 12 shows the validation results for one test with speaker S4 (85.03% accuracy). This is an overlapping speaker, and so the system was trained with similar data. When compared with the single speaker model confusion matrix in Figure 9, we can see that there are some differences. For example, the S5 model confuses ‘soon’ and ‘red’, which the multi person model does not. However, there are many similarities. Figure 12 shows that the model is generally very accurate, with many of the mistakes due to some key errors. The mix up here comes from ‘soon’ and ‘with’, which accounts for a lot of the error, as does a misclassification between ‘now’ and ‘green’. However, apart from these, the results are generally very good, showing that with overlapping speakers, the multi speaker model functions as well as a single speaker model.

However, when we compare the model with non-overlapping speakers, i.e., speakers that the model has not been trained with, we can see that the results vary widely by speaker, as shown in Table 9. We can see this in Figure 13, which is the confusion matrix for one unseen speaker (S12), with an overall recognition rate of 66.57%.

The first thing to note is that the results are worse, as might be expected for an unseen speaker. However, the accuracy of many words is very high. For example, ‘now’ is classified correctly over 80% of the time, as is ‘bin’. There are noticeable classification problems present, for example, ‘soon’ is only classified correctly 28.9% of the time, with it often being predicted as ‘place’. Similarly, ‘blue’ is often predicted as ‘in’, and ‘white’. There are more errors than the with the overlapping speaker confusion matrix shown in Figure 12, but the lower overall score is primarily a result of very poor performance with specific words, rather than a consistent failure.

Finally, we also checked the confusion matrix with another unseen speaker that reported significantly worse performance. The results for Speaker 13 with the same model were much worse (36.3%), as shown in Figure 14. Immediately, it can be seen that the classification is poorer throughout, with the model not correctly able to distinguish between classes, and excessively (and incorrectly) predicting certain words such as ‘bin’ and ‘again’. Conversely, it never correctly predicts ‘soon’, and very rarely predicts ‘set’, ‘lay’, and ‘red’. This shows that unlike the other examples shown with the same model, rather than the error coming from a small number of distinct misclassifications, the model is simply not able to reliably distinguish between classes for this speaker. The three different confusion matrices show that different speakers perform differently with the model. It is possible that creating a larger model with a bigger dataset would allow for better generalisation, but this is not a given.

## 9. Discussion

For the single-person model, different proportions of training data and validation data were used in 2 groups (4 people). Without using a pre-trained model, the average recognition rate was above 80%. After increasing to two persons, the average recognition rate remains above 80%, and increasing to a 6 person model increased the training time, but the recognition rate remained consistently high with overlapping speakers, showing that this method can be extended to cover more speakers without major issues.

As covered in Section 2, and in Table 1, there are a wide variety of methods, training/test conditions, corpora, and tasks, which can make direct comparisons difficult. In terms of direct comparisons with individual word recognition results, Chung and Zisserman [23] achieved 61.1% with overlapping speakers and a CNN, but used a different corpus. Chung et al. used a CNN for training, and a different corpus (LRW) and achieved a recognition rate of 76.20%. Stafylakis and Tzimiropoulos [26] and Petridis et al. [10] also used the same corpus and achieved similar results as those reported in this paper, by using a 3D-CNN and a pretrained CNN (ResNet), although they used the LRW corpus, which splits in broadcast order, so the level of overlap is not always clear.

In terms of comparisons with other results from the GRID corpus, the most notable results are those of Assael et al. with their LipNet approach [15], with 95.20% accuracy. However, it should be noted that they use a big data approach, with an entire overlapping database, and work on a sentence based level. This is solving a different type of problem. Other big data based approaches using CNNs include Chung et al. [16] (97% accuracy with a CNN and LSTM approach), and Xu et al. [21] (97.1% accuracy with a CNN and Bi-GRU approach). Again, these are big data approaches, trained with overlapping speakers and the entire Grid corpus, showing that they learned to recognise the corpus.

Wand et al. [19] reported the results of several different approaches, such as using Eigenlips and a Support Vector Machine (SVM) to get 69.5% accuracy, histogram of oriented gradients (HOG) features and an SVM (71.2%), and Feed-Forward and an LSTM (79.5%), with overlapping speakers. In a later work, Wand et al. [20] used feed-forward and an LSTM network to achieve results of 84.7%, broadly in line with the overlapping speakers reported here.

While our results improve on results by others, such as Wand et al. [20], we note that many of the results in the literature have better results (90%+ accuracy), However, these architectures are very different. Some of these architectures need training on the entire GRID database, such as LipNet [15], or use the huge BBC dataset [16]. These techniques use various CNN models, such as those by Assael et al. [15], Chung and Zisserman [24] and Wand et al. [20]. Their architectures are much more complex, with very large pre-trained models. The results here are achieved using a much smaller dataset, with simple feature extraction that does not require any training, and the speech recognition method used is a bidirectional LSTM model, which again is not pre-trained. This means these results can be achieved much more quickly. In addition to being quick and lightweight, we can also visualise the features over time, and we can identify peaks and troughs, allowing them play a role in justifying and explaining the decisions reached by the system. Thus, our features are intuitive when visualised, compared to less intuitive features. The important contribution of this paper is not only the results, but that good results can be achieved with lightweight and explainable features.

We also considered the mean training times, as shown in Table 10, for a single speaker model, a 2 speaker model, and a 6 speaker model. The results show that with our approach, we can train a single speaker model within 2 h, but that adding extra speakers increases training time substantially. However, part of this is due to the use of Matlab and using the CPU for machine learning, rather than dedicated GPU programming, and future research will aim to optimise this. However, this still compares well to some deep learning models, which are trained for days at a time.

Another issue that was identified was that the speech recognition results do not fully generalise to unknown speakers. However, many approaches in the literature have similar issues, with examples such as LipNet [15] being shown to be completely unusable with a new corpus. This limitation is not uncommon, especially with pixel-based CNN techniques, and it is hoped that further work will result in the development of more generalised approaches that are suitable for real world implementation.

## 10. Conclusions and Future Directions

We presented a very lightweight and quick approach to generating 3 dimensional lip features that can represent words in a way that can be distinctly and consistently visualised (and explained to non domain experts), and can be applied to a wide number of different speakers. However, as with similar approaches, there are limitations, such as problems with facial hair and head turning. As there is no model used, the tracking can easily recover from short term errors. The advantage of our features is that key differences between words can be identified visually and easily. To thoroughly evaluate our proposed approach, we used bidirectional LSTMs to perform visual speech recognition. With trained speakers, the results were more than 80% accurate, similar to those in the literature, with the same caveats as a lot of other research reported in the literature. When testing with unseen speakers from the GRID corpus, a model trained with data from six speakers and tested with nine unseen speakers, gave accuracy ranging between 34% and 63%. Thus, the model has some generalisation ability, but not sufficient, an issue common to other approaches in the literature. Overall, rather than a big data based approach with very large and slow pre-trained models, our approach is human-centric, lightweight and fast, with minimal calibration needed and a simple bidirectional LSTM speech recognition model. Despite this, the results are extremely positive, and show that this approach is effective and reliable. Having demonstrated the effectiveness of our Gabor-based features for speech recognition, this work suggests considering more challenging speech recognition with audiovisual fusion, frame based audio speech estimation, frame based speech estimation [7], and speech filtering [59].

## Figures and Tables

**Figure 1 entropy-22-01367-f001:**
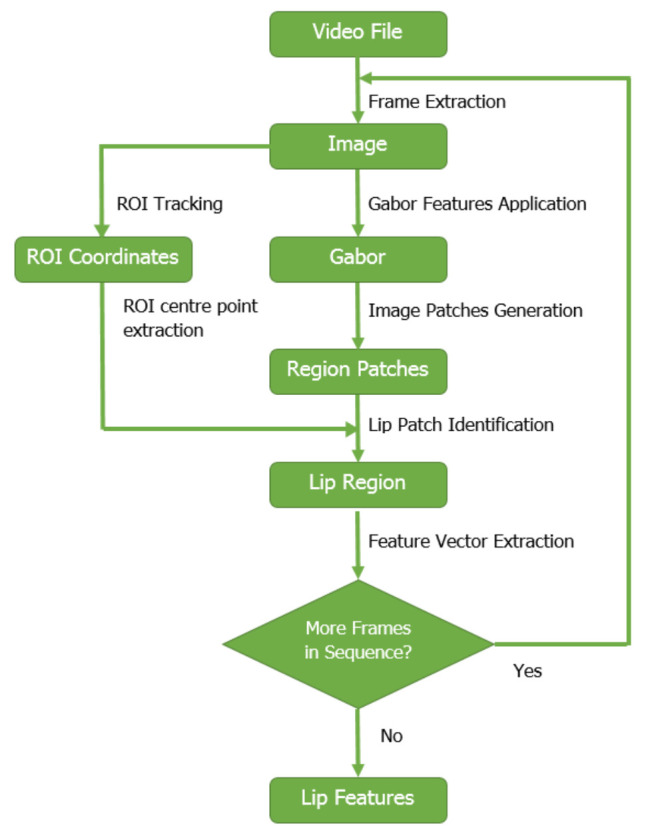
Key stages of lip feature extraction, also reported in [17].

**Figure 2 entropy-22-01367-f002:**
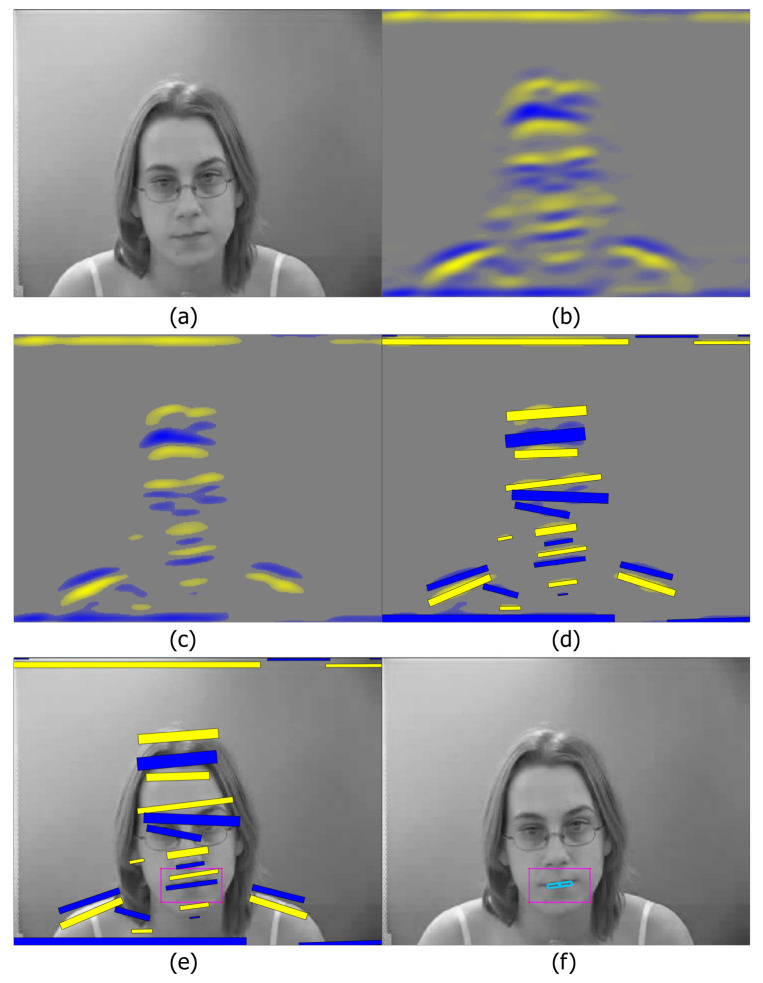
The lip patch generation process, showing (**a**) the original greyscale image, prior to processing, (**b**) the real component of the Gabor features, (**c**) the thresholded image, (**d**) the resulting Gabor image patches, (**e**) the image patches and the tracked ROI box for that frame, and (**f**) the final chosen lip patch Also reported in [17].

**Figure 3 entropy-22-01367-f003:**
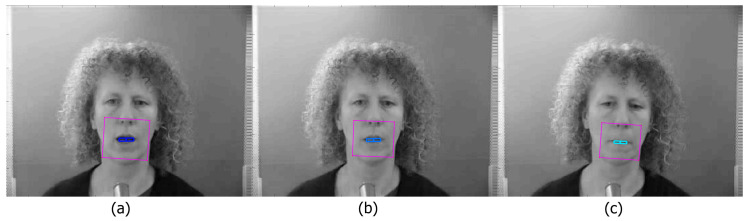
(**a**) Example of a wide open mouth, with a large box and dark colour, (**b**) an open mouth, but without depth (due to the teeth being visible), reflected by the lighter colour, (**c**) a closed mouth.

**Figure 4 entropy-22-01367-f004:**
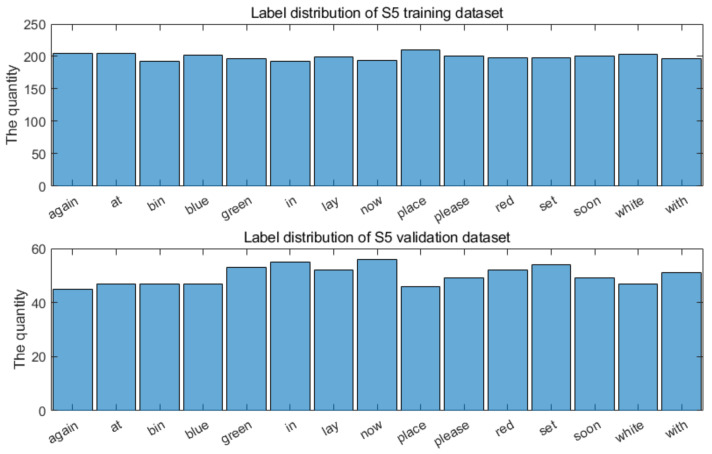
Distribution of words in training/validation sets for a single speaker.

**Figure 5 entropy-22-01367-f005:**
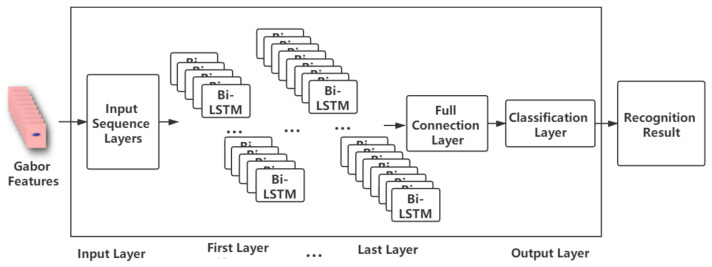
Diagram of speech recognition model, showing the input layer, the hidden layers, and then the fully connected and classification output layers.

**Figure 6 entropy-22-01367-f006:**
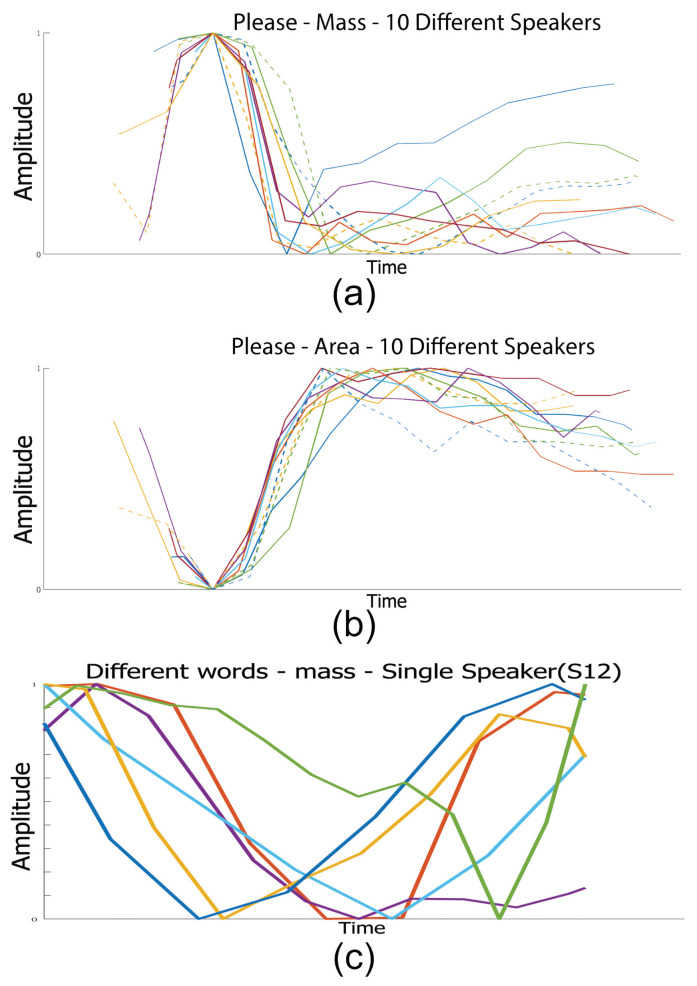
10 normalised sentences from 10 different Grid speakers showing (**a**) mass and (**b**) area for the word ’please’. (**c**) shows the mass for different words from the same speaker for 6 different normalised words, showing at (blue), bin (red), blue (yellow), nine (purple), now (green), q (light blue).

**Figure 7 entropy-22-01367-f007:**
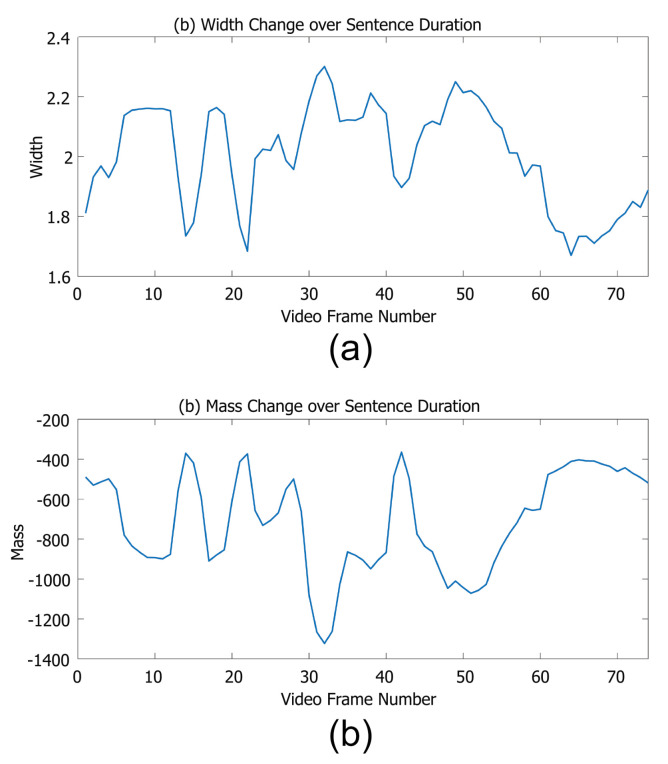
An example of one sentence from the Grid corpus, showing that a sentence can be visualised over time, showing exactly how mouth movement changes over time. We demonstrate here with both (**a**) the width parameter, and (**b**) the mass (the 3d property) parameter.

**Figure 8 entropy-22-01367-f008:**
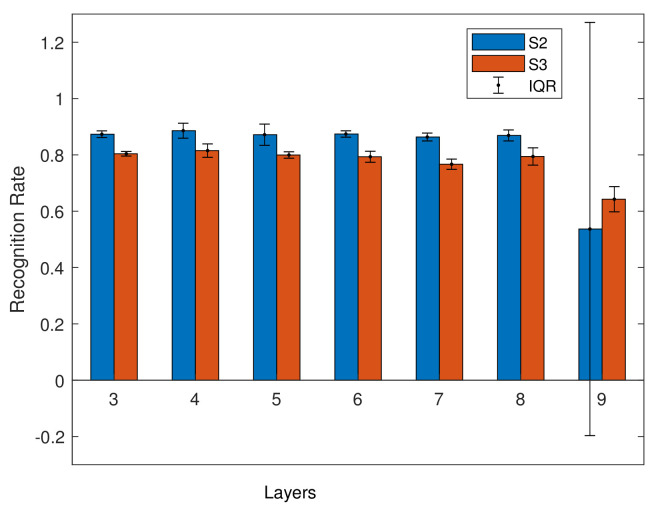
Average recognition rate of S2 and S3 models.

**Figure 9 entropy-22-01367-f009:**
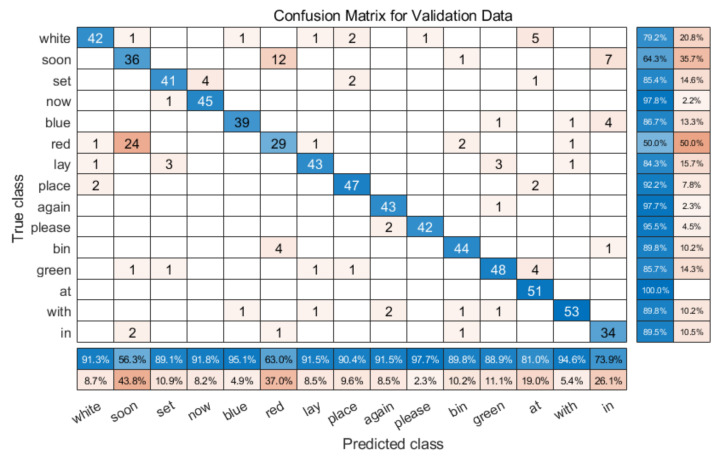
Confusion matrix validation results for single speaker model trained with Speaker 5.

**Figure 10 entropy-22-01367-f010:**
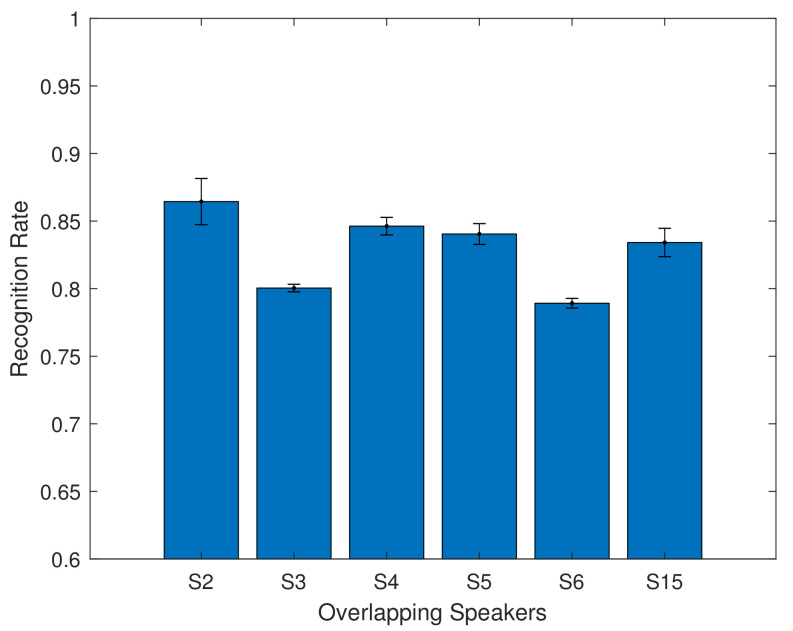
Validation results of overlapping speakers using the multi speaker 6 layers model.

**Figure 11 entropy-22-01367-f011:**
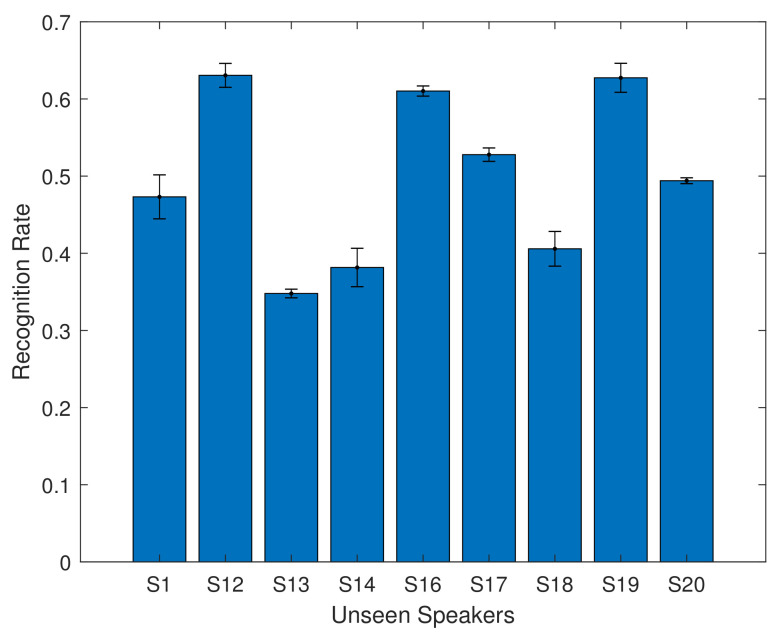
Validation results of unseen speakers, using the 6 layer multi speaker model.

**Figure 12 entropy-22-01367-f012:**
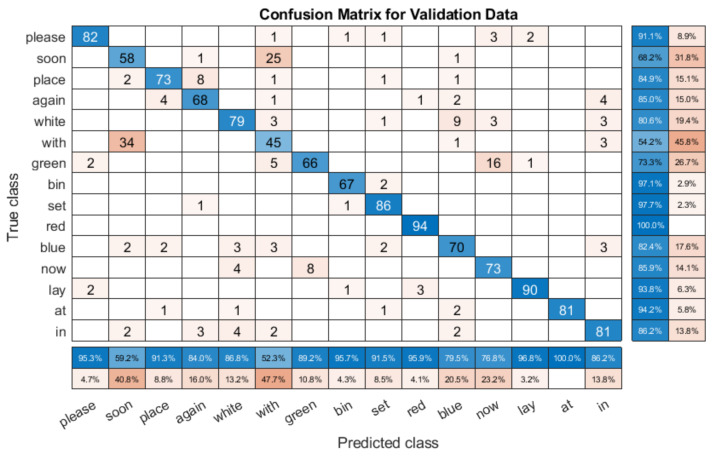
Confusion matrix for multi speaker model with the overlapping S4 validation subset.

**Figure 13 entropy-22-01367-f013:**
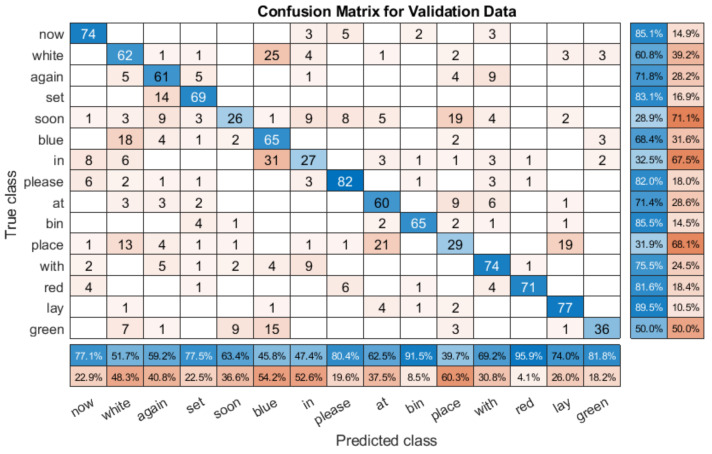
Confusion matrix for the multi speaker model with the unseen S12 validation subset.

**Figure 14 entropy-22-01367-f014:**
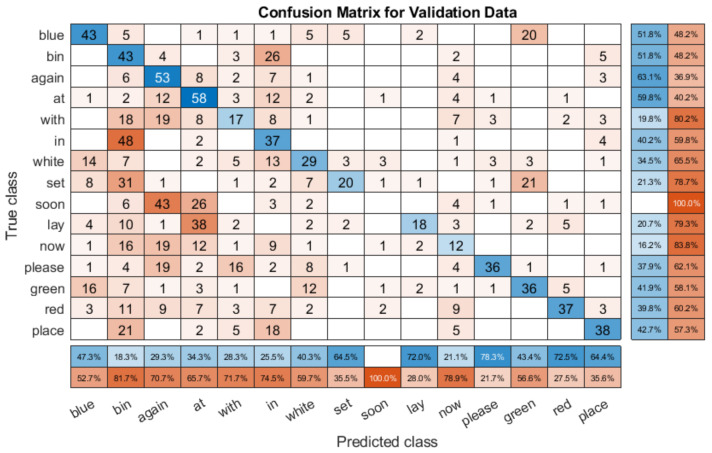
Confusion matrix for multi speaker model with the unseen S13 validation subset.

**Table 1 entropy-22-01367-t001:** Existing lipreading methods using deep learning. There are two key components to traditional lipreading: lip feature extraction (front-end) and feature recognition (back-end). There are also end-to-end systems that use deep learning methods to obtain state-of-the-art performance. For each system, we report the main database, the recognition task tested, and their reported best recognition rate.

Year	Reference	Methods		Database	Recognition Task	Rec. Rate (%)
		Front-End	Back-End			
2016	Assael et al. [15]	3D-CNN	Bi-GRU	GRID	Sentences	95.20
2016	Chung and Zisserman [22]	VGG-M	LSTM	OuluVS2	Phrases	31.90
		SyncNet	LSTM	OuluVS2	Phrases	94.10
2016	Chung and Zisserman [23]	CNN		LRW	Words	61.10
		CNN		OuluVS	Phrases	91.40
		CNN		OuluVS2	Phrases	93.20
2016	Wand et al. [19]	Eigenlips	SVM	GRID	Phrases	69.50
		HOG	SVM	GRID	Phrases	71.20
		Feed-forward	LSTM	GRID	Phrases	79.50
2017	Chung and Zisserman [24]	CNN	LSTM + attention	OuluVS2	Phrases	91.10
		CNN	LSTM + attention	MV-LRS	Sentences	43.60
2017	Chung et al. [16]	CNN	LSTM+attention	LRW	Words	76.20
		CNN	LSTM + attention	GRID	Phrases	97.00
		CNN	LSTM + attention	LRS	Sentences	49.80
2017	Petridis et al. [25]	Autoencoder	Bi-LSTM	OuluVS2	Phrases	94.70
2017	Stafylakis and Tizimiropoulos [26]	3D-CNN + ResNet	Bi-LSTM	LRW	Words	83.00
2018	Fung and Mak [9]	3D-CNN	Bi-LSTM	OuluVS2	Phrases	87.60
2018	Petridis et al. [10]	3D-CNN + ResNet	Bi-GRU	LRW	Words	82.00
2018	Wand et al. [20]	Feed-forward	LSTM	GRID	Phrases	84.70
2018	Xu et al. [21]	3D-CNN+highway	Bi-GRU + attention	GRID	Phrases	97.10
2019	Weng [27]	Two-Stream 3D—CNN	Bi-GUR	LRW	Words	82.07

**Table 2 entropy-22-01367-t002:** Selected feature extraction methods, giving year, the feature extraction method, how speech classification was performed (if used), the database used for classification, and the task carried out, as well as the reported recognition rate

Year	Reference	Feat. Extract.	Classif.	Database	Task	Performance
1988	Kass et al. [32]	ACM				
1998	Cootes et al. [8]	AAM				
2008	Shao and Barker [33]	DCT	HMM	GRID	Phrases	58.40
2008	Seymour et al. [34]	DCT	HMM	XM2VTS	Digits	87.89
		PCA	HMM	XM2VTS	Digits	86.57
		LDA	HMM	XM2VTS	Digits	86.35
2009	Zhao et al. [35]	LBP-TOP	SVM	AVLetters	Alphabet	62.80
		LBP-TOP	SVM	OuluVS	Phrases	62.40
2009	Lan et al. [36]	AAM	HMM	GRID	Phrases	65.00
2009	Dakin et al. [37]	GWT				
2011	Hursig et al. [38]	GWT				
2011	Cappelletta and Harte [39]	Optical flow	HMM	VIDTIMIT	Sentences	57.00
		PCA	HMM	VIDTIMIT	Sentences	60.10
2016	Lee et al. [40]	DCT + PCA	HMM	OuluVS2	Phrases	63.00

**Table 3 entropy-22-01367-t003:** Bi-LSTM model configurations, showing number of neurons in each layer.

Model	Layers								
	1	2	3	4	5	6	7	8	9
3-layers	112	120	128						
4-layers	112	120	120	128					
5-layers	112	112	120	120	128				
6-layers	112	112	120	120	128	128			
7-layers	112	112	120	120	128	128	136		
8-layers	112	112	120	120	128	128	128	136	
9-layers	112	112	120	120	120	128	128	128	136

**Table 4 entropy-22-01367-t004:** Average processing time (5 runs), for S3 dataset single speaker model, varying number of LSTM layers.

LSTM Layers	Mean Training Time (Minutes)	IQR
3-layers	41.75	14.62
4-layers	62.95	18.32
5-layers	78.40	17.25
6-layers	88.07	20.77
7-layers	103.40	11.94
8-layers	139.12	45.23
9-layers	138.80	61.53

**Table 5 entropy-22-01367-t005:** Average recognition rate of all layers for single speaker models S2 and S3.

**Person**	**3 Layers**		**4 Layers**		**5 Layers**		**6 Layers**	
	**Mean**	**IQR**	**Mean**	**IQR**	**Mean**	**IQR**	**Mean**	**IQR**
s2	0.873	0.012	0.886	0.027	0.872	0.038	0.874	0.011
s3	0.807	0.008	0.815	0.024	0.800	0.011	0.793	0.020
**Person**	**7 Layers**		**8 Layers**		**9 Layers**		**Average Value**	
	**Mean**	**IQR**	**Mean**	**IQR**	**Mean**	**IQR**	**Mean**	**IQR**
s2	0.863	0.014	0.869	0.020	0.537	0.734	0.838	0.040
s3	0.767	0.018	0.794	0.031	0.643	0.045	0.778	0.035

**Table 6 entropy-22-01367-t006:** Average recognition rate of S4 and S5 models with single speaker models.

Person	Five Times Tests (Recognition Rate)					Average Recognition Rate
	1	2	3	4	5	
S4	0.851	0.866	0.869	0.873	0.877	0.867
S5	0.871	0.860	0.851	0.870	0.860	0.862

**Table 7 entropy-22-01367-t007:** Average recognition rate of S3 and S4 models with 2 person model.

Person	Five Times Tests (Recognition Rate)					Average Recognition Rate
	1	2	3	4	5	
S3	0.794	0.817	0.813	0.798	0.805	0.805
S4	0.867	0.847	0.834	0.869	0.859	0.855

**Table 8 entropy-22-01367-t008:** Validation results of overlapping speakers using the multi speaker model.

Person	s2	s3	s4	s5	s6	s15
Recognition rate	0.864	0.800	0.846	0.840	0.789	0.834
IQR	0.017	0.003	0.007	0.008	0.004	0.011

**Table 9 entropy-22-01367-t009:** Validation results of unseen speakers with the multi speaker model.

Person	s1	s12	s13	s14	s16	s17	s18	s19	s20
Recog_rate	0.473	0.630	0.347	0.381	0.610	0.527	0.405	0.627	0.494
IQR	0.029	0.016	0.006	0.025	0.007	0.009	0.023	0.019	0.004

**Table 10 entropy-22-01367-t010:** Average processing time for a single speaker model, a dual speaker model, and a six speaker model.

Model	Mean Training Time (Minutes)	IQR
Single Speaker	120.20	15.75
Two Speakers	322.19	138.92
Multi (6) Speakers	1258.50	173.67

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
