# Peer review of "Visual Speech Recognition with Lightweight Psychologically Motivated Gabor Features"

_entropy, 2020, doi:10.3390/e22121367_

Round 1

Reviewer 1 Report

This paper presents an interesting application. Specially due to the interpretability of extracted parameters by humans

I would appreciate some lines explaining the “3D geometric features”. Especially considering that you are working on 2D images, rather than, for instance, time-of-flight images with depth information in addition to (x, y).

You claim that DCT is sensible to illumination. I think the DC component of DCT is sensible to illumination but this is not true for the other coefficients. In addition, I cannot appreciate illumination changes in figures 2 and 3. Thus, I don’t think illumination changes is an issue in this study. If this is an issue, I think you should perform specific experiments to proof this invariance in illumination provided by your method and to illustrate the degradation in recognition accuracies experimented by the other algorithms (for instance, forcing different illumination conditions in training and testing samples).

I wonder if training/testing conditions are the same for all the references of tables [1] and [2]. Probably not, and in this case, direct comparison is not straightforward. If these conditions are different, the tables would benefit if a new column is included with some details about training/testing conditions in each paper.

Line 232: It contains many speakers -> please, provide quantitative information

Some lines/comparison about computational burden of different algorithms would benefit the paper. Is your method suitable for real time? Does it require a very powerful computer or it could be executed in a smartphone?

Author Response

Please see attachment for response to both reviewers.

Reviewer 2 Report

The authors have proposed Visual Speech Recognition system which they claim is lightweight. There have been many visual speech recognition research been done using lip movement features. The authors used a benchmark data set for the evaluation of their approach.

Below are my comments

Author should add some lines about motivation of lightweight approach, maybe for addressing privacy issues, low resource-cost devices and for embodied systems.  

I am missing a computational complexity of all the methods and the proposed approach. It is good if the authors at-least provide their system computational complexity or time to process the test data with their machine specification. 

The comparison of different methods is in background section. I think it is  better to have them in discussion section with your results and the strengths (i.e. pretrained /not pretrained model, speaker dependent/independent, signal processing based features/deep learning based features, computational complexity (if possible)).

Below are some minor concerns: 

1) a native review is needed, the text need polishing and editing

2) Figure 1. -- > no arrow sign for ROI coordinates

3) Figure 5. and 6 etc --> please name figure a, b, c, and add the caption for each separately.

4) better to add a confusion matrix for "the unseen speakers, the average recognition rate is 50.32%."

5) you should also comment on how you will be detecting the start and end of a word, maybe using a visual VAD.

Author Response

(The authors gave the same response as above.)

Round 2

Reviewer 1 Report

The authors improved the paper and it can be accepted as is

Reviewer 2 Report

Thanks for the revised version, I have only one comment about the way you mentioned references in table 1. It is better to follow the same reference style as you used for table 2.